# Pg-GAT: A Complete Graph Model for Cancer Detection and Subtyping in Whole Slide Images Analysis

## Abstract

Whole-Slide-Images (WSIs) have generated significant interests in cancer research community, owing to their availability and the rich information that they provide. Previous Multiple Instance Learning (MIL) methods often neglect the topological structure of tissues which is closely related to tumor evolution. Some attempts with transformer-based MIL methods take spatial relation into account with a trade-off of computational complexity. We propose **P**rojection-**g**ated **G**raph **A**ttention Ne**t**work (Pg-GAT), a lightweight model that effectively leverages graph neural network to provide structural prior, learns spatial and contextual relations through graph attention, and mitigates tissue morphology redundancy with differentiable projection-gated pooling, maintaining a data-adaptive decision boundary. In addition, Pg-GAT outputs region-of-interest (ROI) with respect to the graph-level prediction with post-hoc graph explainer, offering tumor localization and model interpretability. We evaluate our method on lymph node metastasis datasets (CAMELYON16 and CAMELYON17) and non-small cell lung cancer (TCGA-NSCLC), achieving AUCs of 97.6% and 95.6% and 99.6% respectively, outperforming state-of-the-art methods. Code is avaliable at `https://gitlab.com/FUTURE_LINK`

## 1 Introduction

The growing availability of Whole-Slide-Images (WSIs) is transforming the field of digital pathology. However, due to the gigapixel resolution of WSIs, manual annotation and analysis remain prohibitively time-consuming. Recent advancements in artificial intelligence (AI) have enabled significant progress in automating WSI analysis, with multiple instance learning (MIL) being the key paradigm for whole-slide-level analysis. MIL approaches divide WSIs into smaller patches, which are then further analysed via convolutional neural networks (CNNs). However, conventional MIL methods often treat all patches from a WSI as instances within a "bag" and assign a positive label to the entire bag based on the presence of a single positive patch, overlooking important contextual and spatial dependencies between patches.

Attention-based MIL methods are proposed to tackle the missing contextual information problem by learning patch level attention based on extracted patch features. AB-MIL (Ilse et al., 2018) learns the attention of each patch with respect to the slide-level classification. CLAM (Lu et al., 2021) extends AB-MIL with an extra patch clustering branch with pseudo patch label generated by the attention model. Similarly DS-MIL (Li et al., 2021) incorporates a branch of max-pooling to identify critical patches along side the patch attention learning branch. CAMIL (Fourkioti et al., 2023) introduces a context-aware neighbor-constrained mask which is a static 1-hop similarity weighted adjacency matrix in graph construction. However, these methods still overlook the spatial relationships between patches, which are crucial for accurate tissue profiling.

Several other researches leverage transformers (Vaswani, 2017) to incorporate spatial information with positional encoding. TransMIL (Shao et al., 2021) creates artificial 2D square feature map as positional encoding with zero-padding during squaring process, introducing extra non-informative input and alters the tissue spatial structure representation. GTP (Zheng et al., 2022a) constructs a graph with the 2D locations of patches and applys a transformer block with graph adjacency matrices

as positional encoding. This method employs a single layer of graph convolution network (GCN) layer, followed by a computationally expensive min-cut pooling, hindering the scalability.

We observe several limitations in existing spatial-aware and context-aware MIL methods: a) they reply on spectral graphs, requiring extra storage for large adjacency matrix. b) they fail to fully leverage the potential of attention mechanism within graph models. Instead computationally expensive transformer is often adopted. Motivated by this, we seek to explore the efficacy of modern spatial graph models with integrated attention mechanism for WSIs analysis.

In this paper we propose a novel framework for WSIs analysis, namely **P**rojection-**g**ate **G**raph **A**ttention Ne**t**work (Pg-GAT), by exploiting graph structure with attention and empirically chosen n-hop neighborhood. We argue that graph intrinsic characteristics is capable of capturing spatial relations in tissue regions. By incorporating differentiable projection-gated topk pooling in a hierarchical manner, our model efficiently removes morphology redundancy and offers multi-resolution field of view (FOV). Besides being computationally lightweight, Pg-GAT also demonstrates the WSI representation learning efficacy on three benchmark dataset. Furthermore, Pg-GAT can identify tumor regions, offering model interpretability with post-hoc GNNExplainer.

## 2 RELATED WORK

Graph-based WSIs representation learning can be broadly categorized into two approaches: cell-based and patch-based. Cell-based methods (Pati et al., 2022; Nair et al., 2022; di Villaforesta et al., 2023; Alzoubi et al., 2024) rely on precise cell segmentation and effective cell-level feature extraction, which introduces extra uncertainty during preprocessing. In contrast, patch-based approaches offer increased robustness. For instance, GTP (Zheng et al., 2022a) constructs a graph with patches as nodes, connecting them based on Euclidean distance, utilizing a single graph convolutional network (GCN) layer followed by a transformer block that incorporates the graph adjacency matrix as positional encoding. Similarly, CAMIL (Fourkioti et al., 2023) employs a neighbor-constrained matrix that functions as a static 1-hop neighbor similarity matrix in the graph domain. Both methods require substantial storage for large adjacency matrices due to dense matrix multiplication.

Graph pooling is a critical operation for aggregating node-level information to the graph level. In MIL, the class distribution of patches within a single whole-slide image (WSI) is often imbalanced, and traditional pooling techniques such as global mean, max, or sum pooling can lead to undesirable shifts in the decision boundary. Differentiable pooling methods enable graph neural networks (GNNs) to learn the distribution of node classes via backpropagation. DiffPool (Ying et al., 2018) computes soft cluster assignments to coarsen nodes into clusters at each layer. Min-cut pooling (Bianchi et al., 2020), based on the graph min-cut problem, also performs clustering for pooling. Both methods, however, operate on adjacency matrices, resulting in significant storage overhead. More computationally efficient alternatives exist, such as TopK pooling (Gao & Ji, 2019), which uses a 1D projection of node features as a gating criterion, and SAGPool (Lee et al., 2019), which replaces the 1D projection with a GNN layer. ASAP (Ranjan et al., 2020) further extends this by scoring nodes with a GCN after an initial node clustering, incorporating node aggregation and edge weights into the pooling process. However, there is functional overlap between pooling methods like SAGPool, ASAP, and GNNs themselves, which may not necessarily lead to improved model performance.

In contrast to convolutional neural networks (CNNs), where a single convolution operation is often applicable to all image data, the graph domain exhibits greater structural flexibility, making it challenging to design a universal GNN architecture. However, with a deep understanding of GNN architectures and key engineering principles—such as message passing, graph pooling, and domain-specific data characteristics, straightforward and effective GNN models can therefore be developed with recent advances in graph research.

In our proposed Pg-GAT, local information is captured through attention-guided message passing, while global information is aggregated using differentiable graph pooling, addressing the class imbalance problem in WSIs by incorporating both spatial and contextual relationships. Unlike some existing MIL approaches, which apply hierarchical aggregation across multiple image resolutions at the input stage, Pg-GAT performs in-graph hierarchical aggregation. This ensures that the entire process remains intrinsic to the graph representation.

## 3 METHODOLOGY

Pg-GAT operates on the principle that graphs inherently capture spatial relations, while message passing facilitates context exchange among neighboring nodes. Graph pooling mechanisms enable the extraction of global relations. The overall framework of Pg-GAT is depicted in Figure 1.

1. WSIs are divided into non-overlapping patches. A grid graph is constructed with each patch as a node. Node features are extracted via a feature encoder and edges are formed based on the Euclidean coordinates. This initial grid graph serves as the **structural prior**.

2. Pg-GAT takes grid graphs as inputs. Graph attention mechanism learns the interaction importance between nodes/patches with respect to the graph label. Node features are updated via message passing, enhancing the **contextual awareness in local neighborhood**.

3. Projection-gated topk pooling step projects each node feature onto 1D vector with a learnable projection vector $\mathbf{p}$. Projection score $\mathbf{y}$ serves as the gating criterion, leading to the removal of redundant nodes and their corresponding edges after each pooling operation. This step is crucial for learning the **global structure**.

4. Global pooling is applied at each hierarchy, and the slide representation is subsequently aggregated and fed into a classifier for final prediction.

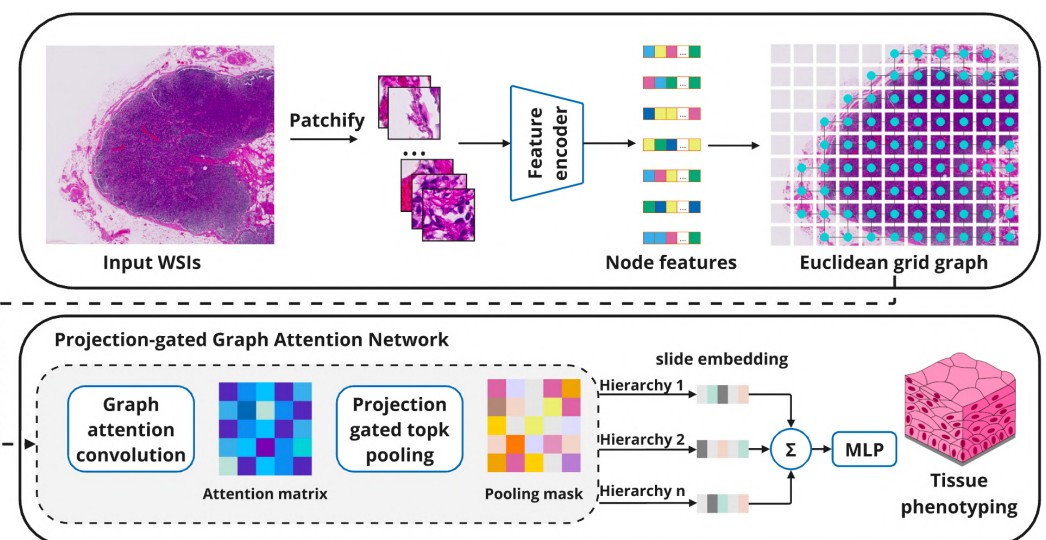

Figure 1: **The Pg-GAT framework**. WSIs are divided into patches with tissue thresholding, and patch features are extracted through a pre-trained feature encoder. A grid graph is constructed with patches as nodes and edges based on Euclidean proximity. Pg-GAT processes the grid graph, learning attention weights between edges and pooling nodes via learnable projection parameters. The slide-level embedding is aggregated through in-graph hierarchical pooling, and tissue phenotypes are classified using a multi-layer perceptron (MLP).

### 3.1 WSI PREPROCESSING AND FEATURE EXTRACTION

We divide WSIs in to $224 \times 224$ non-overlapping patches, with Otsu tissue thresholding at 20x magnification level. To learn meaningful image feature in a self-supervised manner, we utilize discriminate self-supervised pre-training (DINOv2) (Oquab et al., 2023). One pair of augmented views of the query image are sent to a teacher network $g_{\theta_t}$ and a student network $g_{\theta_s}$ which share the same architecture consisting of a backbone ViT (Dosovitskiy, 2020) but different parameters $\theta_t$ and $\theta_s$. Augmentation includes global crop, masked global crop, and local crop. The learning is achieved by minizing the cross entropy of the output of teacher model and student model $P_t$ and $P_s$, which are normalized probability distribution of $K$ dimensions with a temperature parameter $\tau$.

$$P_s(x)^i = \frac{\exp(g_{\theta_s}(x)^{(i)}/\tau_s)}{\sum_{\kappa=1}^{K} \exp(g_{\theta_s}(x)^{(k)}/\tau_s)} \tag{1}$$

$$\mathcal{L}_{DINO} = -\sum P_t log P_s \tag{2}$$

Simultaneously, another iBOT (Zhou et al., 2021) branch learns the sub-patch-level objective with masked global crop.

$$\mathcal{L}_{iBOT} = -\sum P_{ti} log P_{si} \tag{3}$$

The parameters of the student network are optimized via backpropagation, while the teacher network parameters are updated using an exponential moving average of the past iterations. The trained network serves as the feature encoder in Figure 1, generating the initial node embeddings for the constructed graph, denoted as $\mathbf{h}^0 \in \mathbb{R}^{N \times d}$, where $N$ is the number of patches, and $d$ represents the feature dimension of each patch.

## 3.2 PROJECTION-GATED GRAPH ATTENTION NETWORK

Our key assumption is that the intrinsic properties of graphs are more effective and elegant in handling the spatial structure of WSIs compared to positional encodings used in transformers. Given the highly imbalanced distribution of patch classes within a WSI, pooling is crucial for capturing the correct global landscape in slide-level predictions. Projection-gated topk pooling introduces sparsity and hierarchy among the nodes, which we argue is effective in removing morphological redundancies and aggregating data-adaptive, skewed global information.

### 3.2.1 CONTEXTUAL MESSAGE PASSING WITH STRUCTURAL PRIOR

Unlike position encoding in transformer-based methods (Shao et al., 2021; Zheng et al., 2022b;a; Ding et al., 2023), graph naturally captures positional relationships with node connectivity and message passing. The initial grid graph serves as the structural prior. Let $\mathcal{G} = (\mathcal{V}, \mathcal{E})$ be an undirected graph, where $\mathcal{V}$ represents nodes corresponding to patches, and node features given by $\mathbf{h} \in \mathbb{R}^{N \times d}$, with $N$ as the number of nodes and $d$ the feature dimension. Edges are represented by $\mathcal{E}$, where $\mathcal{E}_{i,j} = 1$ if node $i$ and node $j$ are connected. Node interactions are learnt with attention mechanism (Brody et al., 2021) with respect to graph-level prediction. A scoring function $e$ calculates the attention of each neighbor node $j$ for node $i$, with learnable weights $\mathbf{W} \in \mathbb{R}^{d \times d'}$ and attention $\boldsymbol{\alpha} \in \mathbb{R}^{N \times N}$:

$$e(\mathbf{h}_i, \mathbf{h}_j) = \boldsymbol{\alpha}^{\top} \text{LeakyReLU}(\mathbf{W} \cdot [\mathbf{h}_i \parallel \mathbf{h}_j]) \tag{4}$$

The attention scores are then normalized across its neighborhood with Softmax function.

$$\alpha_{ij} = \text{Softmax}_j(e(\mathbf{h}_i, \mathbf{h}_j)) = \frac{\exp(e(\mathbf{h}_i, \mathbf{h}_j))}{\sum_{j' \in \mathcal{N}(i)} \exp(e(\mathbf{h}_i, \mathbf{h}_{j'}))} \tag{5}$$

Node features are updated through message passing, incorporating learnt attention coefficients and weights.

$$\mathbf{h}'_i = \sigma \left( \sum_{j \in \mathcal{N}(i)} \alpha_{ij} \mathbf{W} * \mathbf{h}_j \right) \tag{6}$$

This process enhances the contextual information within the local neighborhood, amplifying features of nodes with higher initial similarity and spatial proximity, while diluting those with lower feature similarity but close proximity. This mechanism facilitates the identification of tumor boundaries.

### 3.2.2 Adaptive Global Structure Learning

We employ projection-gated topk pooling (Cangea et al., 2018; Gao & Ji, 2019), which enables the model to select the most relevant nodes for graph-level predictions, leading to a data-adaptive decision boundary that addresses the imbalanced node classes problem. Node features $\mathbf{h} \in R^{N \times d}$ are projected onto a 1D vector $\mathbf{y} \in R^{N \times 1}$ with a learnable vector $\mathbf{p}$. Top $k$ nodes are chosen after ranking, followed by $tanh$ activation function.

$$\mathbf{y} = \frac{\mathbf{hp}}{\| \mathbf{p} \|} \tag{7}$$

$$\mathbf{i} = \text{top-}k(\mathbf{y}, k) \tag{8}$$

$$\mathbf{h}_{pool} = (\mathbf{h} \odot tanh(\mathbf{y}))_{\mathbf{i}} \tag{9}$$

Here, $\odot$ represents element-wise matrix multiplication, and $\mathbf{i}$ is the index of pooled nodes. After topk pooling, number of nodes is reduced from N to M, $\mathbf{h} \in \mathbb{R}^{N \times d} \rightarrow \mathbf{h}_{pool} \in \mathbb{R}^{M \times d}$, where $M < N$.

Under the grid graph formulation, graph structure is constrained to 8-node connectivity pattern. Nodes share similar degrees. Therefore there is less flexibility compared to more complex graphs such as those in protein structures or social networks. As we show in the results section, projecting node features onto 1D scalar values as pooling gating criterion is an efficient way to learn a meaningful projection direction in pathology WSIs domain.

### 3.2.3 Feature Aggregation and Global Readout

To mimic the varying levels of detail observed by pathologists at different magnifications of WSIs, we aggregate information across hierarchical levels. Unlike prior works such as EGT (Ding et al., 2023), STEMIL (Zhao et al., 2022), and HITP (Chen et al., 2022), which extract features at multiple resolutions during the input stage, we perform in-graph hierarchical aggregation, where node information is aggregated at each level of the learned global structure.

At each graph layer, we apply max pooling following graph attention convolution and topk pooling. For the $l$-th layer, we denote $N^l$ nodes with features $\mathbf{h}^l$. The global graph readout is computed as:

$$\mathbf{h}_G = \frac{1}{L} \sum_{l=1}^{L} \max_{i=1}^{N^l} \mathbf{h}^l \tag{10}$$

The resulting graph-level representation is then fed into an MLP for classification. The depth of the graph neural network, $L$, determines FOV. Hierarchical aggregation enables the capture of both fine-grained information and long-range interactions.

## 4 Experiments and Results

We evaluate our method on three public WSI datasets, CAMELYON16 (Bejnordi et al., 2017)and CAMELYON17 (Bandi et al., 2018) dataest for cancer detection and tumor localization, and TCGA-NSCLC dataset for lung cancer subtyping. Details about the datasets can be found in appendix A.2.

### 4.1 Classification

We present the classification performance using accuracy and Area Under the Curve (AUC) metrics, shown in Table 1. Our primary benchmarks are graph-based models, GTP and CAMIL, alongside non-graph, attention-based models, TransMIL and CLAM. On lung cancer subtyping TCGA-NSCLC dataset, Pg-GAT outperforms the baseline models by a large margin, highlighting the effectiveness of our model in context understanding. On larger dataset CAMELYON16 and CAMELYON17, GTP and CAMIL encounter out-of-memory (OOM) issues due to the need for storing large adjacency matrices for dense matrix operations. CLAM performs relatively well on cancer

Table 1: Classification results on CAMELYON16, CAMELYON17 & TCGA-NSCLC.

| Methods | CAMELYON16 | | CAMELYON17 | | TCGA-NSCLC | |
|---|---|---|---|---|---|---|
| | Acc ($\uparrow$) | AUC ($\uparrow$) | Acc ($\uparrow$) | AUC ($\uparrow$) | Acc ($\uparrow$) | AUC ($\uparrow$) |
| CLAM-SB | $0.930_{0.056}$ | $\mathbf{0.989}_{0.005}$ | $0.924_{0.027}$ | $0.945_{0.015}$ | $0.862_{0.035}$ | $0.937_{0.025}$ |
| CLAM-MB | $\mathbf{0.965}_{0.016}$ | $0.984_{0.007}$ | $\mathbf{0.940}_{0.022}$ | $0.933_{0.029}$ | $0.856_{0.036}$ | $0.939_{0.022}$ |
| TransMIL | $0.871_{0.183}$ | $0.898_{0.151}$ | $0.920_{0.028}$ | $0.950_{0.019}$ | $0.838_{0.009}$ | $0.896_{0.009}$ |
| GTP | OOM[*] | OOM[*] | OOM[*] | OOM[*] | $0.750_{0.024}$ | $0.836_{0.057}$ |
| CAMIL | OOM[*] | OOM[*] | OOM[*] | OOM[*] | $0.838_{0.037}$ | $0.916_{0.032}$ |
| **Pg-GAT** | $0.959_{0.015}$ | $0.976_{0.005}$ | $0.930_{0.002}$ | $\mathbf{0.956}_{0.002}$ | $\mathbf{0.966}_{0.019}$ | $\mathbf{0.996}_{0.004}$ |

[*] We experience OOM but we include the results reported in CAMIL (Fourkioti et al., 2023) for reference. On CAMELYON16, GTP achieves $0.883_{0.026}$ ACC and $0.921_{0.026}$ AUC, while CAMIL achieves $0.917_{0.006}$ ACC and $0.959_{0.001}$ AUC. On CAMELYON17, GTP achieves $0.800_{0.037}$ ACC and $0.762_{0.108}$ AUC, whereas CAMIL achieves $0.843_{0.024}$ ACC and $0.881_{0.039}$ AUC. Note that they used a different feature encoder.

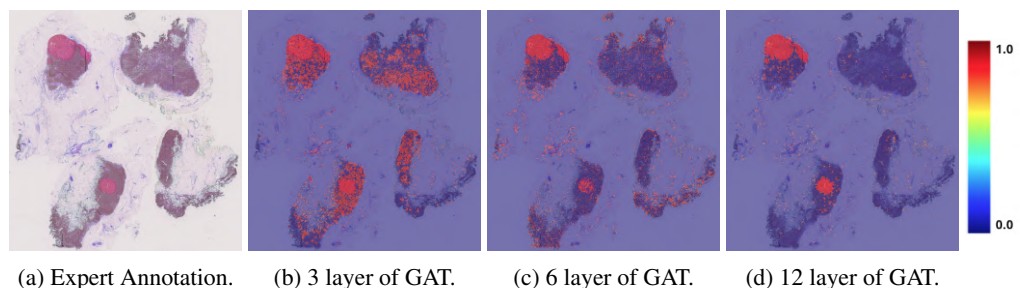

(a) Expert Annotation.  (b) 3 layer of GAT.  (c) 6 layer of GAT.  (d) 12 layer of GAT.

Figure 2: Large tumor region localization. Deeper GNN is better at capturing global dependency, removing sub region level noise.

detection dataset CAMELYON16 and CAMELYON17. Pg-GAT still achieves the highest AUC on the more challenging CAMELYON17 dataset. Notably, cancer subtyping requires a deeper understanding of tumor context compared to tumor/non-tumor classification. The patch clustering branch in CLAM contributes to the tumor/non-tumor detection, but lacks the ability to understand broader context between tumor tissues, thus CLAM underperforms on TCGA-NSCLC dataset compared to our model Pg-GAT. Our Pg-GAT model surpasses baseline methods especially on TCGA-NSCLC dataset, highlighting its strength in performing clinically relevant and context-aware analysis.

## 4.2 TUMOR LOCALIZATION

With trained model, we utilize model agnostic GNNExplainer (Ying et al., 2019), which maximizes the mutual information (MI) between a GNN's prediction and distribution of possible subgraph structures, to analyze the interpretability of our model. The prediction of trained GNN model is defined as $Y = \Phi(G, X)$, determined by graph structure $G$ and node features $X$. To find a subgraph $G_s \subseteq G$ and associated node features $X_s$ that maximize the $MI$, the optimization objective is defined as:

$$\max_{G_s} MI(Y, (G_s, X_s)) = H(Y) - H(Y|G = G_s, X = X_s) \tag{11}$$

where $H(*)$ denotes the entropy.

We demonstrate that our model is capable of localizing large tumor region as well as small tumor region in WSIs in Figure 2 3. As shown in Figure 2, deeper graphs can locate the tumor region with less noise in sub regions due to the longer range of message passing.

Following CAMIL (Fourkioti et al., 2023), we report the Dice score for tumor slides and specificity for non-tumor slides in Table 2. For baseline models, we quote the Dice scores provided in CAMIL due to time constraints. However, since tumor localization in our setting is not framed as

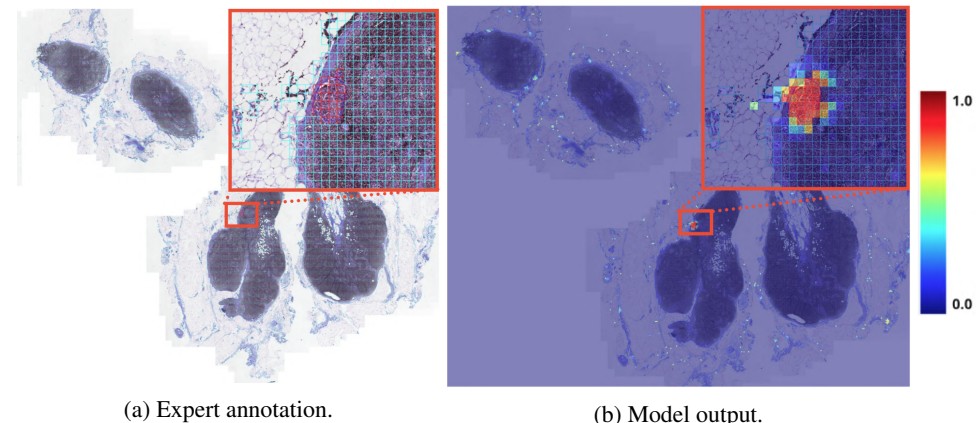

(a) Expert annotation.

(b) Model output.

Figure 3: Small tumor region localization.

Table 2: Tumor localization on CAMELYON16.

| Method | Dice ($\uparrow$) | Specificity ($\uparrow$) |
|---|---|---|
| CLAM-SB | $0.459_{0.037}$ | $0.987_{0.008}$ |
| CLAM-MB | $0.406_{0.007}$ | $0.573_{0.045}$ |
| TansMIL | $0.103_{0.004}$ | $\mathbf{0.999}_{0.001}$ |
| GTP | $0.418_{0.068}$ | $0.851_{0.116}$ |
| DSMIL | $0.259_{0.083}$ | $0.863_{0.043}$ |
| CAMIL | $\mathbf{0.515}_{0.058}$ | $0.980_{0.040}$ |
| **Pg-GAT** | $0.226_{0.006}$ | $0.995_{0.000}$ |

a segmentation task, the Dice score may not serve as the most appropriate metric, and all models demonstrate suboptimal Dice scores. We include it here for reference but did not prioritize its use in our evaluation. We also include more tumor localization visualization in the appendix.

## 4.3 MODEL EFFICIENCY

Our graph-based model, adhering to the principle of Occam's razor, achieves clinically relevant results in WSIs analysis while maintaining architectural simplicity. As shown in Table 3, our model has 17 times fewer parameters than TransMIL. While GTP does not significantly increase parameter count, it requires additional memory for storing adjacency matrices. CAMIL not only requires this extra storage but is also a substantially larger model. Non-graph method CLAM is five times larger than ours. As analysed in (Blakely et al., 2021), as a sparse graph model, Pg-GAT has $\mathcal{O}\left(LEF + LNF^2\right)$ time complexity and $\mathcal{O}\left(LE + LF^2 + LNF\right)$ space complexity. GTP and CAMIL are dense graph model with $\mathcal{O}\left(LN^2F + LNF^2\right)$ time complexity and $\mathcal{O}\left(N^2 + LF^2 + LNF\right)$ space complexity, with $L$ being the number of layers, $E$ the number of edges, $N$ the number of nodes, $F$ the feature dimension. For simplification, we assume the feature dimension remains the same in the next layer. Figure 4 provides an intuitive comparison of model parameter size and AUC performance. Notably, Pg-GAT with 6 and 12 graph attention layers is visualized, having 0.174M and 0.226M parameters with corresponding AUCs of 0.991 and 0.990, respectively, highlighting the model's efficiency even with increased depth.

## 5 ABLATION STUDY

We perform an ablation study by replacing the graph attention layer with a graph convolution layer (GCN). Results are shown in Table 4. Additionally, we evaluate the model with an alternative differential pooling method, SAG Pooling (Lee et al., 2019), which replaces the 1D projection in topk pooling with a GCN layer. A non-differentiable alternative, mean pooling, is also evaluated. Results are presented in Table 5. Our ablation studies indicate that GCN consistently underperforms

Table 3: Model efficiency comparison.

| Method | #Params($\downarrow$) | Graph Computation |
|--------|-----------------------|-------------------|
| CLAM-SB | 0.791M | Non-graph |
| CLAM-MB | 0.792M | Non-graph |
| TansMIL | 2.672M | Non-graph |
| GTP | 0.172M | Dense |
| CAMIL | 1.871M | Dense |
| **Pg-GAT** | 0.149M | Sparse |

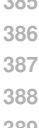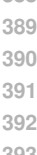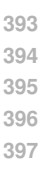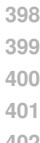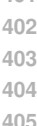

Figure 4: Model comparison on TCGA-NSCLC dataset. Each bubble's area is proportional to parameter size of a variant in a model family. CLAM sub-family includes CLAM-SB and CLAM-MB. Pg-GAT family includes 3-layer, 6-layer and 12-layer of graph attention layers.

compared to GAT. SAG Pooling does not offer performance improvements and introduces higher computational costs, while mean pooling fails to capture the adaptive global structure. The results on CAMELYON17 dataset exhibit a greater discrepancy due to the more pronounced patch class imbalance compared to the TCGA-NSCLC dataset. These observations underscore the role of the attention mechanism in understanding local neighborhood context, while projection-gated topk pooling is sufficient in learning meaningful graph pooling criterion. Together, these components are crucial for capturing both spatial- and context-awareness, enabling the learning of adaptive global structures.

## 6 CONCLUSION

In this work, we proposed Pg-GAT, a novel graph-based framework for WSI analysis that incorporates spatial- and context-awareness with in-graph hierarchical aggregation, emulating the decision-making process of pathologists. Pg-GAT captures node interactions using an initial Euclidean grid graph as a structural prior and enhances contextual awareness within local neighborhoods through graph attention. The differentiable projection-gated pooling mechanism enables the model to learn data-adaptive decision boundaries, which is particularly important in handling imbalanced class distributions typical in the WSIs domain. We demonstrated the effectiveness of our approach on three benchmark datasets using accuracy and AUC metrics, offering model interpretability with tumor localization, as well as its computational efficiency through model complexity analysis.

Table 4: Graph convolution layers comparison.

| Methods | CAMELYON16 | | TCGA-NSCLC | |
|---------|------------|------------|------------|------------|
| | Acc ($\uparrow$) | AUC ($\uparrow$) | Acc ($\uparrow$) | AUC ($\uparrow$) |
| **Pg-GAT** | $\mathbf{0.959}_{0.015}$ | $\mathbf{0.976}_{0.005}$ | $\mathbf{0.968}_{0.019}$ | $\mathbf{0.996}_{0.007}$ |
| **Pg-GCN** | $0.950_{0.013}$ | $0.960_{0.008}$ | $0.966_{0.012}$ | $0.994_{0.000}$ |

Table 5: Graph pooling comparison.

| Methods | CAMELYON17 | | TCGA-NSCLC | |
|---------|------------|------------|------------|------------|
| | Acc ($\uparrow$) | AUC ($\uparrow$) | Acc ($\uparrow$) | AUC ($\uparrow$) |
| **Pg-GAT** | $\mathbf{0.930}_{0.002}$ | $\mathbf{0.956}_{0.002}$ | $0.967_{0.008}$ | $\mathbf{0.996}_{0.000}$ |
| **SAG-GAT** | $0.911_{0.003}$ | $0.942_{0.002}$ | $\mathbf{0.970}_{0.012}$ | $\mathbf{0.996}_{0.000}$ |
| **Mean-GAT** | $0.816_{0.008}$ | $0.877_{0.003}$ | $0.954_{0.033}$ | $0.983_{0.000}$ |

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

# A   APPENDIX

## A.1   REPRODUCIBILITY STATEMENT

We use PyTorch (2.2.2), NVIDIA RTXA5000. One GPU is used for each training. We intend to make our code publicly available.

## A.2   DATASET

CAMELYON16 (Bejnordi et al., 2017) and CAMELYON17 (Bandi et al., 2018) are breast cancer lymph node metastasis dataset, both with lesion level annotation by pathologists. For CAMELYON17, there are also lesion sub-level labels:

- macro: Metastases greater than 2.0 mm
- micro: Metastases greater than 0.2 mm or more than 200 cells but smaller than 2.0 mm
- itc: Isolated tumor cells. Single tumour cells or a cluster of tumour cells less than 0.2 mm or fewer than 200 cells are not precisely a metastasis but are instead classified as single tumour cells or a cluster of tumour cells smaller than 0.2 mm or less than 200 cells

Classifying a whole slide as a tumor slide becomes more challenging when the tumor region is confined to very small sub-level areas.

TCGA-NSCLC lung cancer dataset, from The Cancer Genome Atlas Program, consists of two types lung cancer, lung adenocarcinoma (LUAD) and lung squamous cell carcinoma (LUSC), but does not have tumor region annotation.

## A.3   DATA SPLITS

We perform 5-fold cross validation, 270 samples with 80% train and 20% validation split and 129 test samples from official grand challenge on CAMELYON16. On CAMELYON17, we perform 4-fold cross validation, 506 samples with 70% train 15% validation and 15% test split with the same sub-level lesion label distribution. On TCGA-NSCLC we perform 5-fold cross validation, 920 samples with 70% train 15% validation and 15% test split. All with standalone test set. CAMELYON dataset is with experts annotations of tumor regions, thus is used for downstream ROI investigation.

### A.3.1   WSI PREPROCESSING FEATURE EXTRACTION

We adopt the standard WSIs preprocessing (Lu et al., 2021), segmenting the tissue region with Otsu thresholding, then dividing the remaining images into none-overlapping $224 \times 224$ patches. To minimize computational overhead and take advantage of the rich feature representations acquired from prior training, we utilize UNI (Chen et al., 2024) pathology foundation model, which utilizes self-supervised learning DINOv2 (Oquab et al., 2023) for pathology slide feature learning, and is not trained on public dataset CAMElYON16, CAMELYON17 and TCGA, thus there is no data leakage risk in our evaluation. Same feature encoder is applied to all experiments.

## A.4   MORE EXAMPLES OF TUMOR LOCALIZATION

### A.4.1   GOOD CASES

We first present more good cases in Figure 5 6 and non cancerous examples in Figure 7.

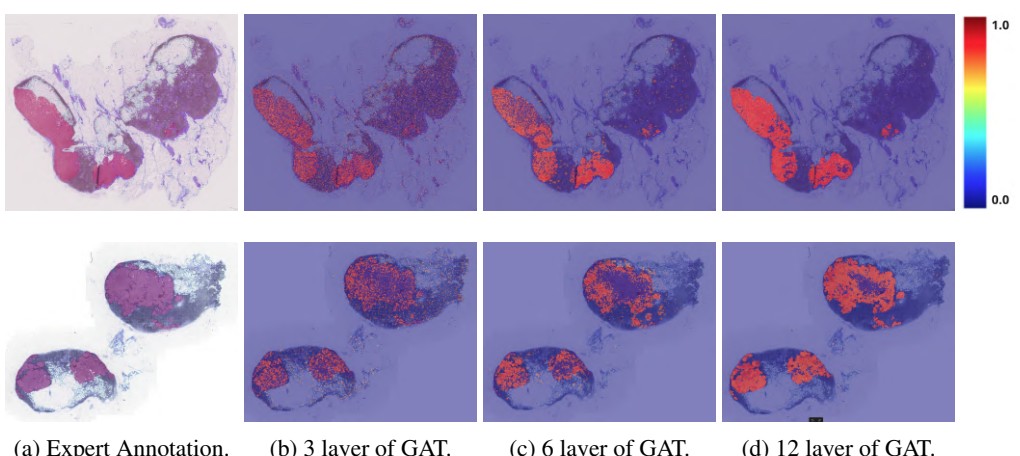

(a) Expert Annotation.    (b) 3 layer of GAT.    (c) 6 layer of GAT.    (d) 12 layer of GAT.

Figure 5: Tumor localization. Deeper GNN is better at capturing global dependency, removing sub region level noise.

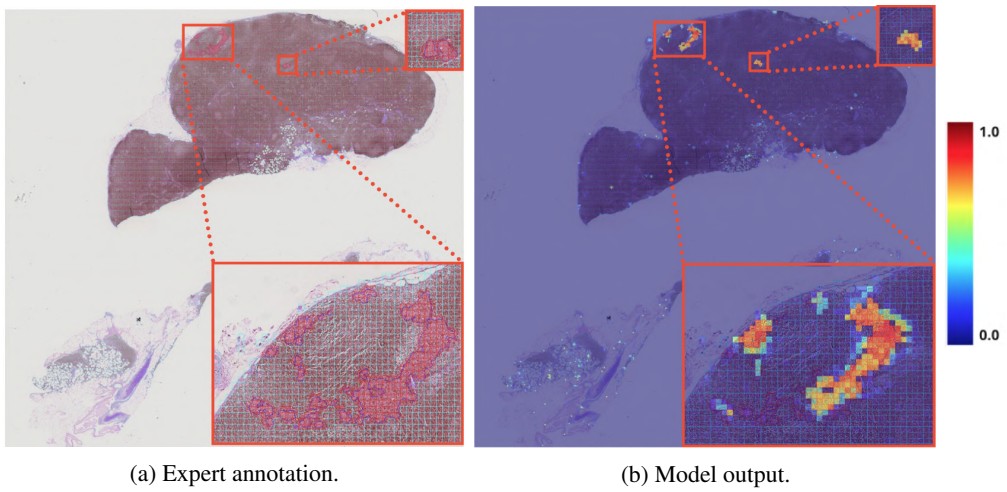

(a) Expert annotation.                    (b) Model output.

Figure 6: Small tumor region localization.

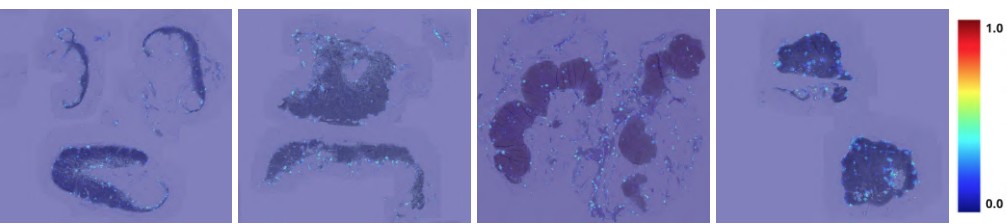

Figure 7: Non cancerous slides.

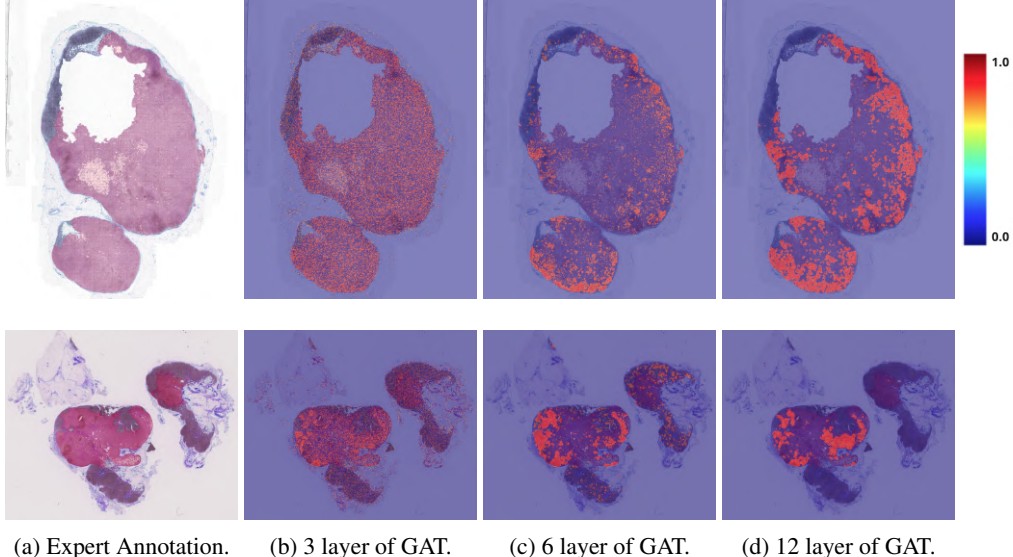

| (a) Expert Annotation. | (b) 3 layer of GAT. | (c) 6 layer of GAT. | (d) 12 layer of GAT. |

Figure 8: Failure cases. 12-hop is too small for this large tumor region case.

### A.4.2 FAILURE CASES

Here we also present failure cases in Figure 8. We notice 12-hop is too small for very large tumor region cases. The depth of GNN is a hyperparameter, the further tuning of which was limited by time constrains.

