# OpenReview forum: "Pg-GAT: A Complete Graph Model for Cancer Detection and Subtyping in Whole Slide Images Analysis"
_ICLR.cc/2025/Conference — Submitted to ICLR 2025_

### Official Review · Reviewer_SeyR · 2024-10-18

**Soundness:** 3
**Presentation:** 3
**Contribution:** 2
**Rating:** 5
**Confidence:** 4

**Summary:**

This paper propose a lightweight graph network for whole-slide images analysis, in which the nodes are grid patches and edges are initialized as the Euclidean distances. Compared with previous methods, the proposed GAT can efficiently learn both the local context information and global structure that are hard to extract from whole-slide images. More specifically, the projection-gated pooling technique can introduce sparsity and hierarchy among graph nodes, making it effective to remove morphological redundancy and aggregate global information. In overall, the graph formulation for whole-slide images is interesting and straightforward. However, part of the experiment results are not solid enough and do not achieve the SOTA performance. If my concerns can be addressed, I would like to raise my ratings to borderline above.

**Strengths:**

- The graph formulation for whole slide images are straightforward and interesting, and more importantly, it is costly efficient.

- The global structure learning is inspiring. The ways of how to remove morphological redundancy and perform in-graph hierarchical aggregation are effective and non-trivial.

- The paper is well-written, and the method can be easily followed by readers.

**Weaknesses:**

My major concerns are about the experiments:

- The proposed method cannot achieve SOTA performance on Camelyon 16 and 17 benchmarks. Why not listing the challenge winners in table 1? For example, in Camelyon 16 challenge, the winner (Harvard & MIT) already achieves 99.4% AUC. Also, the SOTA of Camelyon 17 is from DeepBio Inc. For more recent results on the two benchmarks, you can check table 1 in PFA-Scannet [MICCAI 2019].  It is suggested to include a comparison to these top-performing methods and explain how your approach compares in terms of performance and computational efficiency.

- Why not using the challenge metrics for evaluation? For Camelyon16, FROC is a more challenging metric compared with AUC reported in this paper. Also, kappa score should be compared for Camelyon17 benchmark. Please explain why you chose AUC over FROC for Camelyon16 and to provide results using both metrics if possible. Similarly, for Camelyon17, results using the kappa score are requested, which would allow for a more direct comparison to other methods evaluated on this benchmark.

- The tumor localization performance is not satisfying. Even though the proposed method is not prioritized for this task, the gap of CAMIL, around 3% dice, is too large. Please discuss potential reasons for this performance gap in tumor localization, and suggest ways that might improve this aspect of proposed method, even if it's not the primary focus.

**Questions:**

- Any justification of using DINOv2 as the pretraining image feature? The recent pretrained RADIO feature can be a better design choice, since it is distilled from CLIP, DINOv2 and SAM, containing richer information (both low-level correspondence and high-level semantics) in the features. Please compare the performance of your method using DINOv2 versus RADIO features, or to explain why you believe DINOv2 is more suitable for your task.

- How much morphological redundancy can be removed after performing top-k ranking? Please. provide quantitative results showing the reduction in node count (from N to M), and to discuss how this affects the model's performance and efficiency.

- The effectiveness of GAT is questionable, compared with GCN. As shown in table 4, the improvement of GAT over GCN is very marginal. Please provide a more detailed analysis of why GAT was chosen over GCN.

**Details Of Ethics Concerns:**

N/A.

---

> ### Author Response · Authors · 2024-11-18
>
> Thank you for the review.
> ### Weaknesses (in the same order)
> - We did not compare with the challenge papers because we are comparing semi-supervised learning (mainly with graphs).  Our training only requires slide-level labels. We believe that’s also why similar works with CAMELYON dataset [1][2][3] didn’t cite challenge results either.
> - Following work[1], we simplify the prediction task to binary classification for the comparison with other baseline models, so we did not compute kappa score for CAMELYON17. We agree that it’s good to include FROC for tumor localization, which we didn’t have time for. We are working on that. AUC is good for WSIs classification.
> - The potential reason could be 1) As shown in the failure cases, our model is missing some part of a very large tumor due to limited tuning in the depth of GAT layers. 2) We construct the edges based on spatial distance and the prediction task is only graph level. We thought about creating a separate branch for node/patch level prediction to enhance the tumor localization, but in this case, the model requires patch level annotation, which is not semi-supervised anymore. But it could be an interesting future work with more annotated data.
>
> ### Questions (in the same order)
> - RADIO is an interesting work. But unfortunately its release timeline does not fit with our project. We were not able to investigate more into that. But after reading a little more about it, RADIO seems to incorporate more language capability. DINOv2 is focused on learning visual features, especially the image retrieval which is interesting in WSI domain, while CLIP and SAM are trained with images and languages together. Also as shown in Figure 5 in RADIO paper, DINOv2 seems to have the best visual representation around 224px for the image which has a similar foreground and background. That is the similar case with H&E staining images. We’d believe that in some sense this supports our choice of DINOv2 for our use case. RADIO could be future work with pathology image description generation.
> - For 3-layer GAT, our configuration is pooling with ratio [0.8,0.6,0.4] for each layer. It is a hyperparameter. We chose this ratio based on experience. It could be optimized with a hyperparameter tuner.
> - We were also a little surprised by the good performance of GCN and we thought it might be because we used a pathology foundation model which offers very good initial node features. Maybe GAT would outperform GCN more if we train end-to-end from scratch, which we do not have the hardware capability to do so. In the end we took the performance gain of GAT as it is.
>
>
> [1] Fourkioti, Olga, et al. "CAMIL: Context-aware multiple instance learning for cancer detection and subtyping in whole slide images." arXiv preprint arXiv:2305.05314 (2023).
>
> [2]Li, Bin, Yin Li, and Kevin W. Eliceiri. "Dual-stream multiple instance learning network for whole slide image classification with self-supervised contrastive learning." Proceedings of the IEEE/CVF conference on computer vision and pattern recognition. 2021.
>
> [3]Bontempo, Gianpaolo, et al. "A Graph-Based Multi-Scale Approach with Knowledge Distillation for WSI Classification." IEEE Transactions on Medical Imaging (2023).

---

> > ### Comment · Reviewer_SeyR · 2024-11-19
> >
> > My major concerns are not fully addressed, e.g., the evaluation metrics such as FROC and kappa score are still not included in the comparison. My other concerns such as RADIO feature, and the justification of GAT over GCN, are not well explained, neither. Therefore, I cannot raise my rating.

---

### Official Review · Reviewer_FhjX · 2024-10-30

**Soundness:** 2
**Presentation:** 3
**Contribution:** 1
**Rating:** 3
**Confidence:** 5

**Summary:**

The paper introduces a Projection-gated Graph Attention Network (Pg-GAT) for Whole-Slide Image (WSI) analysis, focusing on cancer detection and subtyping. Pg-GAT leverages a graph neural network (GNN) framework to model spatial and contextual relationships within tissue structures, advancing traditional Multiple Instance Learning (MIL) methods. With a differentiable pooling mechanism, it aims to reduce tissue morphology redundancy while maintaining interpretability through tumor localization.

**Strengths:**

- Proposes a lightweight GNN architecture that models spatial and contextual relationships within WSIs, claiming efficiency and interpretability.
- Comprehensive experimental setup that benchmarks against notable graph- and non-graph-based models.
- Demonstrates the capacity for tumor localization, with interpretability results provided through visualization and GNNExplainer analysis.

**Weaknesses:**

- **Novelty limitations**: The proposed use of Graph Attention Networks (GAT) for WSI-based cancer analysis has precedent in prior studies, reducing the novelty of Pg-GAT in this context[1,4]. DASMIL [1], for example, already uses a GAT layer and message passing to allow interaction along patches on different resolutions before attention pooling.
- **Performance Concerns**: Results in Tables 1 and 2 do not consistently outperform baselines across all metrics. Specifically, Pg-GAT's tumor localization scores, evaluated by the Dice score, and classification accuracy fall short of fully distinguishing it from competing methods. More relevant confidence-based metrics, such as FROC, are related to localization and Camelyon. Since it is based on confidence, it measures the capabilities to distinguish diseases without any manual threshold.
- **Limited Scope in Model Efficiency Analysis**: Figure 4, presenting model size versus performance, is restricted to a single dataset, limiting insights into Pg-GAT’s comparative efficiency across different dataset complexities.
- **Different backbone**: Attempts to reproduce baselines such as GTP and CAMIL resulted in out-of-memory (OOM) issues, so results were drawn from prior publications. However, this approach introduces limitations, as these reported baselines utilize a different backbone (SIMCRL), which could affect comparability and undermine the experimental rigor. That comparison is vital to quantifying how the solution is better than other GNN-based solutions.
- **Interpretability Limitations**: The interpretability section would benefit from more generalized tumor examples to substantiate that Pg-GAT can adapt to varied tumor complexities.

### Minor Comments
- Clarify abbreviations as Acc and Minor typographical errors in section headers and figure captions should be corrected (e.g., "TansMIL" to "TransMIL").
- Several mentioned methods do not appear in the comparison tables as ABMIL, DSMIL, STEMIL, EGT  **(graph-based)**
- Limited Focus in Related Work: The Related Work section centers primarily on GNN-based approaches for WSI analysis rather than providing a balanced discussion that includes broader Multiple Instance Learning (MIL) solutions for WSIs. A more comprehensive review of MIL approaches used in generic WSI settings would better contextualize Pg-GAT within the landscape of WSI analysis techniques. Additionally, CAMIL and GTP are presented as primary graph-based baselines, but they are not the only approaches that employ graphs for WSI representation( EGT, DASMIL[1], H2MIL[2], GDSMIL[3]). Including a discussion on other graph-based MIL solutions would provide a clearer picture of how Pg-GAT fits among similar models and address potential limitations in novelty.
- This manuscript lacks implementation details (lr, scheduler, epochs, batch_size, etc.), which doesn't allow for reproducibility

[1]: A Graph-Based Multi-Scale Approach With Knowledge Distillation for WSI Classification, TMI
[2] H^2-MIL: Exploring Hierarchical Representation with Heterogeneous Multiple Instance Learning for Whole Slide Image Analysis
[3] Enhancing PFI Prediction with GDS-MIL: A Graph-Based Dual Stream MIL Approach.
[4] Whole Slide Cervical Cancer Screening Using Graph Attention Network and Supervised Contrastive Learning

**Questions:**

Why do you have OOM with other baselines? Are you working with batch_size=1? Is the issue related to training or inference? Consider exploring subgraph sampling during training to address the OOM issues with baselines. This approach might allow for dropping patches while augmenting the number of slides, potentially improving efficiency and performance.  What is the contribution of the work to other GAT-based solutions?

---

### Official Review · Reviewer_r9WZ · 2024-11-02

**Soundness:** 1
**Presentation:** 2
**Contribution:** 1
**Rating:** 3
**Confidence:** 5

**Summary:**

This paper introduces Projection-Gated Graph Attention Network (Pg-GAT) for analyzing whole-slide images (WSIs) in pathology. Pg-GAT leverages graph neural networks to capture structural, spatial, and contextual relationships while reducing redundancy in tissue morphology. The model employs a projection-gated pooling mechanism to create adaptive decision boundaries and provides both tumor detection and localization.

**Strengths:**

1. The projection-gated pooling mechanism is a novel contribution, adding value to the model's adaptability.
2. The paper includes comprehensive experiments at both slide and patch levels to evaluate its effectiveness.

**Weaknesses:**

1. The use of spatial graphs and graph attention networks for WSI analysis can't be viewed as contribution in 2024. Several prior graph-based methods for WSIs, such as PatchGCN [1], TEA-Graph [2], and SlideGraph+ [3], already incorporate spatial encoding and graph neural networks for WSI analysis. This prior work is notably missing from the paper’s introduction, overlooking important context for the field.

2. The paper’s critique of earlier methods lacks specificity. It is unclear which prior methods employ spectral graphs in WSI analysis, or why spatial graphs, as implemented here, would not require a large adjacency matrix. Additionally, the claim of “unleashing the potential of attention mechanisms” is vague; graph attention networks (introduced in 2017) [4] have long supported attention mechanisms, raising questions about Pg-GAT’s unique contributions beyond the projection-gated pooling.

3. Evaluating Pg-GAT on a binary cancer classification task does not fully leverage the model’s contextual learning capabilities, as the task could be solved with only one positive patch. A more suitable evaluation might involve context-dependent tasks, such as survival prediction, where broader spatial relationships are essential.

4. The baselines in Table 1 are insufficient, especially with two methods omitted due to out-of-memory errors, leaving only three comparison models (two of which are variants of CLAM). This limited comparison fails to provide a comprehensive benchmark against relevant methods.

5. The unexpectedly strong performance of Mean-GAT in TCGA-NSCLC compared to more established methods like TransMIL, CLAM, and GTP suggests potential issues in evaluation or methodology. This inconsistency should be clarified to ensure the robustness of Pg-GAT’s reported results.

6. Pg-GAT’s low detection performance in Table 2 contradicts its claim of effective tumor localization. This discrepancy undermines the model’s stated contribution of providing interpretability and detection capabilities.

[1]. Chen, R. J., Lu, M. Y., Shaban, M., Chen, C., Chen, T. Y., Williamson, D. F., & Mahmood, F. (2021). Whole slide images are 2d point clouds: Context-aware survival prediction using patch-based graph convolutional networks. In Medical Image Computing and Computer Assisted Intervention–MICCAI 2021: 24th International Conference, Strasbourg, France, September 27–October 1, 2021, Proceedings, Part VIII 24 (pp. 339-349). Springer International Publishing.

[2]. Lee, Y., Park, J. H., Oh, S., Shin, K., Sun, J., Jung, M., ... & Kwon, S. (2022). Derivation of prognostic contextual histopathological features from whole-slide images of tumours via graph deep learning. Nature Biomedical Engineering, 1-15.

[3]. Lu, W., Toss, M., Dawood, M., Rakha, E., Rajpoot, N., & Minhas, F. (2022). SlideGraph+: Whole slide image level graphs to predict HER2 status in breast cancer. Medical Image Analysis, 80, 102486.

[4]. Veličković, P., Cucurull, G., Casanova, A., Romero, A., Lio, P., & Bengio, Y. (2017). Graph attention networks. arXiv preprint arXiv:1710.10903.

**Questions:**

1. Overall, while Pg-GAT’s projection-gated pooling is an interesting contribution, the paper requires substantial improvement in establishing its novelty, clarifying its claims, and providing comprehensive evaluations to be competitive within the field of computational pathology.
2. The provided GitLab link was inaccessible; please verify the link’s accessibility before including it in the submission.

---

> ### Author Response · Authors · 2024-11-21
>
> 1. a) Our idea is Occam’s razor. A very simple graph method outperforms complex SOTA models and we elaborate on the reasoning.
> b) PatchGCN did not offer results on slide level classification, and no tumor localisation evaluation. It only showed 2 out of 4370 samples of low/high risk attention map. So we left it out. We do not have access to TEA-Graph. SlideGraph+ is a cellular graph, not in our review scope.
> 2. Previous works in the baseline models [1][2] both require sorting adjacency matrics as shown in their paper and code. Works on Graph constitutional network[3][4] reduce the complexity from O(n²) to O(nk) of n nodes and their k neighbors, because it does not require storing large adjacent matrics. That is the foundation of graph convolution networks,. “unleashing the potential of attention mechanisms” means GAT only outperforms complex models mentioned in the paper.
> 3. Survival prediction is more interesting, but we don’t have the data. Baseline models also did slide level classification.
> 4. OOM is an important disadvantage to mention because light weight is necessary in real-life scenarios. If we also quote results directly from relevant works as done in work[5], we would move the footnote numbers into the tables and we outperform by large margins.
> 5. In work[5] tabel II, they also achieved good results on TCGA-NSCLC with mean pooling. We believe it is because of the more balanced patch classes in TCGA-NSCLC slides if you take a look at the slides. It also makes sense that mean pooling works worse on CAMELYON because they have more unbalanced patch classes. We don’t know what other models did wrong to underperform mean pooling.
> 6. We thought about creating a separate branch for node/patch level prediction to enhance the tumor localization, but in this case, the model requires patch level annotation, which is not semi-supervised anymore. But it could be an interesting future work with more annotated data. We first want to show how capable a simple graph model is.
> 7. link inaccessibly: the link is a placeholder, but we would leave it empty to avoid misunderstanding.
>
> [1] Fourkioti, Olga, et al. "CAMIL: Context-aware multiple instance learning for cancer detection and subtyping in whole slide images." arXiv preprint arXiv:2305.05314 (2023).
>
> [2]Zheng, Yi, et al. "A graph-transformer for whole slide image classification." IEEE transactions on medical imaging 41.11 (2022): 3003-3015. NO CAMELYON
>
> [3] Defferrard, Michaël, Xavier Bresson, and Pierre Vandergheynst. "Convolutional neural networks on graphs with fast localized spectral filtering." Advances in neural information processing systems 29 (2016).
>
> [4] Kipf, Thomas N., and Max Welling. "Semi-supervised classification with graph convolutional networks." arXiv preprint arXiv:1609.02907 (2016).
>
> [5] Bontempo, Gianpaolo, et al. "A Graph-Based Multi-Scale Approach with Knowledge Distillation for WSI Classification." IEEE Transactions on Medical Imaging (2023).

---

> > ### Comment · Reviewer_r9WZ · 2024-11-21
> >
> > The rebuttal does not clearly articulate its unique contribution in relation to prior work. Concerns regarding the overall contribution and experimental results remain unaddressed. Therefore, I will maintain my current rating.

---

### Official Review · Reviewer_bBcf · 2024-11-04

**Soundness:** 1
**Presentation:** 1
**Contribution:** 1
**Rating:** 1
**Confidence:** 5

**Summary:**

This paper proposes the Projection-gated Graph Attention Network (Pg-GAT) framework, a hierarchical graph framework composed of successive GAT + Gated Projection TopK Pooling layers. After each GAT + TopK layer, max pooling is applied to obtain a readout vector. The readouts are finally averaged and input into the final MLP classification layer. This approach is applied to three benchmark datasets: CAMELYON16, CAMELYON17 and TCGA-NSCLC and compared to 4 baseline methods (CLAM, TransMIL, GTP, CAMIL). Performant results are shown across datasets. Post-hoc interpretability heatmaps were obtained using the GNNExplainer module. Ablation was carried out on the type of pooling operation (TopK, SAGPool, Mean) and graph convolution (GAT, GCN).

**Strengths:**

The paper is clear and well written.

**Weaknesses:**

This work replicates the pipeline established in MUSTANG [1] published at BMVC 2023, without proper acknowledgment or citation. The fundamental proposed architecture and application area - hierarchical GAT layers combined with pooling for WSI analysis - is identical to MUSTANG's published approach.

The only differences are the authors:

- Use a "grid-graph" defined on spatial nearest neighbors, rather than a k-NN graph defined in feature space. The use of spatial nearest neighbor graphs was established in [2], which is not cited either.
- Use of Projection-gated topK pooling [3] instead of SAGPooling [4]. Ablation on this was also carried out in MUSTANG.
- Apply this approach to benchmark CAMELYON and TCGA datasets, rather than the multi-stain dataset used in MUSTANG.
- Apply pos-thoc GNNExplainer[5]

Neither of these differences represents a significant technical contribution for a conference such as ICLR. I therefore recommend rejection based on lack of novelty and proper attribution.

I have raised further concerns regarding this submission with the AC, SAC and PCs.

[1] Gallagher-Syed et al. Multi-Stain Self-Attention Graph Multiple Instance Learning Pipeline for Histopathology Whole Slide Images. BMVC 2023. https://papers.bmvc2023.org/0789.pdf

[2] Chen et al. Whole Slide Images are 2D Point Clouds: Context-Aware Survival Prediction using Patch-based Graph Convolutional Networks. MICCAI 2021. https://arxiv.org/abs/2107.13048

[3] Cangea et al. Towards Sparse Hierarchical Graph Classifiers. NeurIPS 2018 Relational Representation Learning Workshop. https://arxiv.org/abs/1811.01287

[4] Lee et al. Self-Attention Graph Pooling. ICML 2019. https://arxiv.org/abs/1904.08082

[5] Ying et al. GNNExplainer: Generating Explanations for Graph Neural Networks. NeurIPS 2019. https://arxiv.org/abs/1903.03894

**Questions:**

- How does the pipeline proposed in this work differ from that proposed in MUSTANG?

**Details Of Ethics Concerns:**

I believe this submission does not properly acknowledge or cite work it is based on, which constitutes plagiarism.

---

> ### Author Response · Authors · 2024-11-14
>
> Regarding your plagiarism claim:
> 1. First, we were unaware of the paper in question, aka MUSTANG. It has a limited citation history, with only three citations (one self-citation and one survey), and it was not cited by the papers we referenced.
> 2. There are existing works which used GAT+SAGpool in pathology[1][2] prior to MUSTANG. Playing around with different graph networks and pooling is a standard approach.
> 3. Here we reiterate the differences even you said some of the differences yourself:
>     1. **Graph Construction**: The first step is already different. We constructed the graph based on spatial euclidean distance while the MUSTANG is based on feature similarity KNN. Spatial graph construction is a well-known, intuitive method used across several studies. Given its common use, we did not focus on finding an origin reference for this approach and thus did not consider it necessary to cite.
>     2. **Framework Structure**: Unlike MUSTANG claimed end-to-end method, we employ a two-stage approach, as shown in our flowchart. Our first stage involves self-supervised feature learning to initialize node features in WSI graphs. The second stage involves a GAT combined with a learnable pooling mechanism.
>     3. **Tumor Localization**: We deployed tumor localization, which does not appear to be achievable by MUSTANG. In the MUSTANG paper, the graph visualization does not provide a one-to-one correspondence with WSI patches.
>     4. Ablation studies on other graph networks and pooling methods are also common sense to do.
>
> Even after carefully reviewing MUSTANG now, we feel that citing it is unnecessary due to its limited relevance to our methodology, reasoning, and visualizations. We do not claim novelty of each component. We propose a novel framework that combines well-established components with a clear, empirically supported rationale for its strong performance.
>
> [1] Zuo, Yingli, et al. "Identify consistent imaging genomic biomarkers for characterizing the survival-associated interactions between tumor-infiltrating lymphocytes and tumors." International Conference on Medical Image Computing and Computer-Assisted Intervention. Cham: Springer Nature Switzerland, 2022.
>
> [2] Wang, Zichen, et al. "Hierarchical graph pathomic network for progression free survival prediction." Medical Image Computing and Computer Assisted Intervention–MICCAI 2021: 24th International Conference, Strasbourg, France, September 27–October 1, 2021, Proceedings, Part VIII 24. Springer International Publishing, 2021.

---

> > ### Comment · Reviewer_bBcf · 2024-11-19
> >
> > Dear authors, thank you for your comment. However, given the lack of novelty of the components, as well as the overall framework - and it's poor grounding in relevant graph literature (PatchGCN, H2-MIL, TEA-Graph, SlideGraph, MUSTANG, HEAT, amongst other works) I cannot change my opinion regarding acceptance.

---

### Official Review · Reviewer_emVG · 2024-11-06

**Soundness:** 1
**Presentation:** 3
**Contribution:** 1
**Rating:** 3
**Confidence:** 5

**Summary:**

The paper introduces Pg-GAT, a graph-based framework for whole-slide image (WSI) analysis, enhancing spatial and contextual awareness with in-graph hierarchical aggregation. By using an initial Euclidean grid graph and projection-gated pooling, Pg-GAT effectively adapts to imbalanced class distributions typical in WSIs. The model achieves good performance on benchmarked TCGA datasets. However, The authors seem to lack an understanding of relevant work and advancements in this field, the proposed concept/method is not novel, and I could not see their contributions to this field.

**Strengths:**

- **Clear Logical Flow:** The paper is structured logically, presenting each component of Pg-GAT in a clear, sequential manner that enhances readability and understanding.

- **Organized and Coherent Writing:** The writing is well-organized, with concise explanations that make mentioned concepts more accessible to the reader.

- **Comprehensive Visualizations:** The paper provides visualizations that illustrate the model’s performance and tumor localization capabilities, adding clarity and support to the claims.

**Weaknesses:**

- **Lack of Novelty in Graph Construction:**
The authors appear to lack familiarity with existing work in WSI analysis. Their proposal of leveraging spatial correlations in WSI graphs is not novel at all. For example, H2MIL[1] constructs a hierarchical graph based on spatial relationships, TEA-Graph[2] establishes spatial graphs while incorporating node similarity for graph construction, and HEAT[3] uses heterogeneous graphs based on node relations. The proposed Pg-GAT method does not clearly demonstrate novelty over these established approaches.

- **Insufficient Comparison:**
Although Pg-GAT is evaluated on benchmark datasets, there is a lack of direct comparison with recent methods in WSI analysis, like DTFD-MIL[4], making it difficult to assess its competitive performance in this context.

- **Invalid Code Link:**
The authors put an invalid code link in their abstract, I think they should upload a valid code link.

- **Overclaim of Contribution:**
The authors argue they offer model interpretability, however, they just directly use well-established tools proposed by other papers.

- **Insufficient Evaluation:**
The authors claim their model is computationally lightweight, however, they did not compare the FLOPs in their experiments.

[1] Hou W, Yu L, Lin C, et al. H^ 2-MIL: exploring hierarchical representation with heterogeneous multiple instance learning for whole slide image analysis[C]//Proceedings of the AAAI conference on artificial intelligence. 2022, 36(1): 933-941.

[2] Lee Y, Park J H, Oh S, et al. Derivation of prognostic contextual histopathological features from whole-slide images of tumours via graph deep learning[J]. Nature Biomedical Engineering, 2022: 1-15.

[3] Chan T H, Cendra F J, Ma L, et al. Histopathology whole slide image analysis with heterogeneous graph representation learning[C]//Proceedings of the IEEE/CVF Conference on Computer Vision and Pattern Recognition. 2023: 15661-15670.

[4] Zhang H, Meng Y, Zhao Y, et al. Dtfd-mil: Double-tier feature distillation multiple instance learning for histopathology whole slide image classification[C]//Proceedings of the IEEE/CVF conference on computer vision and pattern recognition. 2022: 18802-18812.

**Questions:**

- **Enhance Novelty in Graph Construction:** To distinguish Pg-GAT from existing methods, consider integrating unique graph construction techniques, such as incorporating both spatial and semantic relationships in node connectivity or exploring heterogeneous graph structures. Highlighting any specific advantages of your approach compared to H2MIL, TEA-Graph, or HEAT would also strengthen the novelty of your work.

- **Expand Experimental Comparison:** Adding comparisons with recent methods like DFTMIL could provide a clearer benchmark for Pg-GAT's performance. Evaluating Pg-GAT against a broader set of state-of-the-art approaches will help establish its competitive strengths and limitations.

- **Update or Clarify Code Link:** Ensure that the code link in the abstract is valid and accessible. If the code is not yet available, consider specifying a release timeline in the abstract or mentioning that it will be provided upon publication.

- **Clarify Interpretability Contributions:** If model interpretability is a key contribution, consider developing additional interpretability techniques beyond standard tools, or clarify the unique insights Pg-GAT offers. Explicitly discussing how your model leverages these tools in a way that adds value could address overclaim concerns.

- **Include Computational Efficiency Metrics:** To substantiate claims of computational efficiency, incorporate FLOPs or runtime comparisons with similar models. This would provide quantitative evidence of Pg-GAT’s computational advantages and validate the claim of being lightweight.

---

### Meta-Review · Area_Chair_dmVL · 2024-12-16

**Metareview:**

This paper proposes a GNN-based model for whole slide image analysis to provide structural prior for contextual modeling. All reviewers raised questions about the lack of novelty, incomplete comparison, and overclaimed contributions. For example, this paper shows poor grounding in relevant graph literature in whole slide image analysis. Moreover, the baseline methods for comparison and metrics for evaluation are limited.

Given the consensus of reviewers, rejection is recommended.

**Additional Comments On Reviewer Discussion:**

Reviewers raised questions and challenged the novelty of methodology, limited experiments, and overclaimed contributions. Although the authors responded to several reviewers, the reviewers consistently agreed that major concerns regarding the overall framework and experiments were not fully addressed.

---

### Decision · Program_Chairs · 2025-01-22

Reject